# Strontium Ranelate and Strontium Chloride Supplementation Influence on Bone Microarchitecture and Bone Turnover Markers—A Preliminary Study

**DOI:** 10.3390/nu16010091

**Published:** 2023-12-27

**Authors:** Karolina Turżańska, Agnieszka Tomczyk-Warunek, Maciej Dobrzyński, Maciej Jarzębski, Rafał Patryn, Joanna Niezbecka-Zając, Monika Wojciechowska, Aneta Mela, Aneta Zarębska-Mróz

**Affiliations:** 1Department of Rehabilitation and Orthopaedics, Medical University of Lublin, Jaczewskiego 8, 20-954 Lublin, Poland; karolinaturzanska@gmail.com (K.T.); asiareh@gazeta.pl (J.N.-Z.); aneta.zarebska-mroz@umlub.pl (A.Z.-M.); 2Department of Pediatric Dentistry and Preclinical Dentistry, Wroclaw Medical University, Krakowska 26, 50-425 Wroclaw, Poland; maciej.dobrzynski@umw.edu.pl; 3Department of Physics and Biophysics, Faculty of Food Science and Nutrition, Poznan University of Life Sciences, Wojska Polskiego 38/42, 60-637 Poznan, Poland; maciej.jarzebski@up.poznan.pl; 4Department of Humanities and Social Medicine, Medical University of Lublin, Chodźki 7, 20-093 Lublin, Poland; rafal.patryn@umlub.pl; 5Department of Pediatrics and Nephrology, Medical University of Lublin, 20-093 Lublin, Poland; monikaw87@op.pl; 6Department of Experimental and Clinical Pharmacology, Medical University of Warsaw, 02-097 Warsaw, Poland; aneta.mela@gmail.com

**Keywords:** strontium ranelate, strontium chloride, growing mice, bone microarchitecture, microtomography

## Abstract

Despite strontium ranelate use in osteoporosis management being one of the promising concepts in disease treatment, there is no clear evidence that strontium organic compounds are more effective than inorganic ones. The aim of this study was to compare strontium chlorate and strontium ranelate influence on the mice bone microarchitecture. We investigated whether strontium chlorate (7.532 mmol/L) and strontium ranelate (7.78 mmol/L) solutions fed to healthy SWISS growing mice (*n* = 42) had an influence on the percent of bone volume (BV/TV), trabecular thickness (Tb.Th), number of trabeculae (Tb.N), and separation between each trabecula (Tb.Sp) in the chosen ROI (region of interest) in the distal metaphysis of the left femurs. The cortical bone surface was examined close to the ROI proximal scan. There was an increase in each examined parameter compared with the control group. There were no statistical differences between strontium ranelate and strontium chlorate parameters. Our study indicates that organic and inorganic strontium compounds similarly affect the bone microarchitecture and strength.

## 1. Introduction

Osteoporosis is a growing worldwide problem with enormous social and economic costs [1]. One possible prevention measure to postpone the disease’s development is building the highest peak bone mass (PBM) and increasing bone quality, e.g., by improving bone microarchitecture. Peak bone mass (the highest bone mass in the lifetime) is achieved in early adulthood, usually mid-twenties. According to the literature, populational changes in PBM have a more significant impact on the development of osteoporosis than the extended period before menopause [2]. Also, it was shown that bone quantity achieved during growth has a more significant influence on delaying osteoporosis than the subsequent process of losing it [3]. The modern lifestyle (lack of physical activity, bad diet, and other factors) can severely influence the quantity of bone tissue achieved as PBM [4]. Per McDevitt, exercise and nutrition are critical modifiable factors for establishing good bone health and optimizing PBM during growth [5]. Karlsson and Rosengren’s 2020 review proved that increasing children’s physical activity level is undoubtedly beneficial for their future PBM [6]. Unfortunately, it has also been proved that modern life tends to be more and more sedentary [7]. 

The research shows that an increase in PBM by only 10% delays the development of osteoporosis by over ten years [8]. Therefore, beyond lifestyle intervention, pharmacological stimulation, which could positively impact the achievement of peak bone mass and maintenance of the quality of bone tissue as long as possible in the lifetime, could be beneficial [9]. 

The positive effect of strontium ions on bone tissue has been known for a long time [10,11]. Numerous studies have demonstrated its beneficial effects on bone quality in animal and cell models of osteoporosis, leading to the development of strontium ranelate as an anti-osteoporotic drug. Strontium ranelate, with its unique “dual agent” effect (inhibiting resorption and stimulating bone formation at the same time), proved to be very efficient in osteoporosis treatment [10]. Unfortunately, some side effects appeared in recent years in people at increased risk of thromboembolism, consequently leading to the withdrawal of the strontium ranelate in 2017 [12] since this formulation of strontium is not available in most markets except for the UK, where the drug came back in generic form in 2019 [13]. Since then, strontium salts have been available on the market in the form of poorly tested supplements containing mainly strontium in the form of chloride or citrate, which are advertised mainly as an alternative to anti-osteoporotic drugs [14]. Considering the positive effect of strontium ranelate on bone tissue [10], the authors decided to compare its action on the bone microarchitectural parameters and bone turnover markers as a standard substance with inorganic strontium salt (strontium chloride). Considering strontium chloride’s potential efficacy as a supplement supporting the development and maintenance of peak bone mass during youth and adulthood, the fast-growing young, intact individuals were chosen as the experimental model.

## 2. Materials and Methods

### 2.1. Animals

The experiment was carried out based on the consent issued by the Local Ethical Committee No. 1 at Lublin Medical University.

Forty-two growing male mice (Swiss, six weeks old, mean body mass 22.3 g (±3.3 g)) were randomly assigned to three groups: (I, *n* = 14; II, *n* = 14; and III, *n* = 14). A power analysis of the test was used to determine the size of the experimental groups, which was carried out using the Statistica program (Version 14.0.0.15 TIBCO Software Inc., Palo Alto, California, U.S.). 

Animals were pair-housed in polycarbonate cages and kept in standard laboratory conditions—light cycle 12/12 h, room temperature 21 ± 3 °C, and humidity 55 ± 5%. All animals were fed a standard laboratory diet (0.5% of total calcium and 0.75% of total phosphorus) ad libitum and had free access to drinking solutions. Controls of group I drank water, whereas strontium was added to drinking water as chloride (7.532 mmol/L) for group II and ranelate (7.78 mmol/L) for group III. During the experiment, body weights were monitored weekly. Amounts of water and food consumed by animals were checked daily. Fresh drinking solutions were prepared every day. After eight weeks, the animals were sacrificed under deep barbiturate anesthesia. Femurs were harvested, stripped of soft tissue, wrapped in phosphate buffer saline-soaked gauze, and deep frozen (−80 °C) until the micro-CT examination. Blood samples were collected and secured for further investigation (bone turnover markers).

### 2.2. Micro-CT Analysis

In order to evaluate the microarchitecture of bone tissue, the area of the distal metaphysis of the left femurs of all animals was subjected to microtomographic examination. After defrosting, the samples were placed on a centric stand in the center of the field of view of the microtomography chamber (Skyscan 1072, SkyScan, Antwerp, Belgium) and scanned (magnification 34×, Pixel 8.42, Rotation step 0.23, Exposure 1.9sec, and Gain 1). All of the samples were meticulously positioned so the mid-shaft of the bones was vertically straight and the orientation of the bones was consistent. Raw scans were reconstructed using the nRecon program (Skyscan, Belgium) and then thoroughly analyzed using the Ctan program (Skyscan, Belgium). An identical ROI (region of interest) was subjected to detailed morphometric evaluation. For the trabecular bone in all samples, a 100-slice cylinder was set in the central part of the distal epiphysis at a visual distance from the cortical bone, at the height of 85% of the bone length (Figure 1 and Figure 2). The analysis started at the most proximal end of the distal growth plate, which was visually identified in subsequent sections and taken as the reference point. In the ROI of each tested sample, the following microarchitectonic parameters were analyzed: percent of bone volume (BV/TV), trabecular thickness (Tb.Th), trabecular number (Tb.N), and trabecular separation (Tb.Sp). The surface of the cortical bone was assessed by subtracting the area whose outer border was the cortex’s inner circumference from the femur’s total surface at the level of the most proximal scan of the ROI from each group. Length measurements were measured from the left femur using a caliper. Representative micro-CT overview images are shown in Figure 3, Figure 4 and Figure 5.

### 2.3. Examination of the Level of Bone Turnover Markers in the Blood Serum

For bone turnover marker (C-terminal telopeptide of type I collagen—CTX I, C-terminal telopeptide of type II collagen—CTX II, and procollagen type I N-terminal propeptide—PINP) assessment, nine randomly selected samples from each group were chosen and measured with rodent Elisa kits (Immunodiagnostic Systems, Boldon, United Kingdom). Only samples that did not undergo hemolysis during preparation were used for analysis. The number of samples was reduced to nine so that the tested groups were homogeneous in size. In the case of CTXI, the internal and external inter-assay variability was <5.8% and <10.7% for blood serum; for CTXII, the internal variability was <5.6% and the external variability was <10.5%; and for P1NP, the internal variability was <5.0% and the external variability was <8.2%.

### 2.4. Statistical Analysis

All statistical analysis was carried out in Statistica Version 14.0.0.15 TIBCO Software Inc. Palo Alto, California, U.S. Data were ensured to be approximately normal using a Shapiro–Wilk test when a one-way ANOVA was used. A Tukey–Kramer post hoc test was used where significance was found. A Kruskal–Wallis ANOVA test was used where data were non-normal, per the Shapiro–Wilk test. Statistical significance was determined when *p* < 0.05. The tables present three levels of statistical significance: * <0.05, ** 0.002, and *** 0.001. 

## 3. Results

### 3.1. General Observations

No deaths were recorded, none of the animals were visibly ill during the experiment, and no anomalies were noted during the macroscopic autopsy of the animals. Fluid and food intakes were similar in all groups and consistent with those presented in the literature [15,16].

### 3.2. Body Mass

There was no difference in weight gain between the first and last day of the experiment (Table 1). One-way ANOVA analysis revealed no significant differences.

### 3.3. Trabecular Micro-Architecture

#### 3.3.1. Trabecular Thickness (Tb.Th), Trabecular Number (Tb.N), and Percent of Bone Volume (BV/TV) 

Regarding trabecular thickness (Tb.Th), the results showed an increase in both studied groups compared with the control group (27% strontium chloride vs. control *p* < 0.5; 37% strontium ranelate vs. control *p* < 0.2). There was no significant difference between the strontium chloride and strontium ranelate groups. For trabecular number (Tb.N), the results showed an increase of 53% for strontium chloride vs. control (*p* < 0.02) and an increase of 112% for strontium ranelate vs. control groups (*p* < 0.001). There was also no significant difference between both examined groups. For percent of bone volume, the results were similar: 101% increase for strontium chloride vs. control (*p* < 0.02) and 185% increase for strontium ranelate vs. control (*p* < 0.001). All calculations were made using one-way ANOVA, followed by the post hoc Tukey–Kramer test. The results are summarized in Table 2, Table 3, Table 4, Table 5, Table 6 and Table 7.

#### 3.3.2. Trabecular Separation

Regarding trabecular separation (Tb.Sp), data were non-normal, per the Shapiro–Wilk test, and a Kruskal–Wallis ANOVA test was used. The results showed a decrease in the separation between trabeculae of 34% in strontium chloride vs. the control (*p* < 0.001) and 44% between strontium ranelate vs. the control (*p* < 0.02). No statistical difference was found between the strontium chloride and the strontium ranelate groups. The results are summarized in Table 8 and Table 9.

### 3.4. Cortical Bone Area

Cortical area analyses showed an increase of 34% in strontium chloride vs. the control (*p* < 0.001) and an increase of 30% in strontium ranelate vs. the control (*p* < 0.02). No significant difference was found between both examined groups. All calculations were made via one-way ANOVA, followed by the post hoc Tukey–Kramer test. The results are summarized in Table 10 and Table 11.

### 3.5. Bone Turnover Markers

One-way ANOVA revealed no significant difference between the control and strontium chloride and the control and strontium ranelate groups, and no significant difference was found between the strontium chloride and strontium ranelate groups. The results are summarized in Table 12, Table 13 and Table 14.

## 4. Discussion

This study was designed as a supplementation experiment. To avoid the stress-inducing effect of the oral gavage, the animals were given constant access to drinkers with solutions of the tested substances instead of water. Those preventive measures were made to limit the negative effects of endogenous catecholamines, both on the quality of bone tissue and the length of animal lifespan. Despite awareness of the limitations resulting from the selected administration model without food restrictions [17], the selected model was considered to be the most similar to the natural model of dietary supplementation. In a study conducted in 2008, the authors proved the effective incorporation of strontium into the bone, both in the form of ranelate and chloride, in a model identical to that used in the current experiment [18].

An appropriate model for experimental animals is a crucial element of any experiment. In the case of anti-osteoporotic drugs, efficacy tests based on the model of animals subjected to ovariectomy are considered optimal [19,20]. In the discussed study, however, the focus was not on the treatment of osteoporosis but on the effectiveness of using strontium chloride as a dietary supplement, which would reverse the impact of modern lifestyle on achieving peak bone mass and maintaining it later in life. Therefore, the model of growing young animals with metabolically active bone tissue was considered the most beneficial. The intact animal model was widely used in strontium ranelate preclinical (animal) studies. In Delannoy’s 2002 two-year study, strontium ranelate was given to healthy male and female mice to check the drug’s long-term influence on vertebral bone metabolism [21]. In 2009, Ammann used 48 intact female rats to prove that oral strontium ranelate improves intrinsic bone quality [22]. In the experiment conducted on 344 six- to seven-week-old intact Fischer rats, the same author proved that strontium ranelate improves bone resistance by increasing bone mass and improving bone architecture. Moreover, in 2013, Wohl used an intact growing male rat’s model to check strontium bone content in the strontium ranelate and strontium citrate groups [23]. The growing mice model was used in the strontium ranelate osteogenesis imperfect study [24], and an intact rat’s model was applied in a study on strontium ranelate influence on condylar growth during mandibular advancements [25].

Strontium is poorly absorbed in the digestive tract. Vitamin D, lactose, and other carbohydrates support this process, while calcium ions inhibit it [26]. Absorbed strontium is almost completely incorporated into the bone [27]. The element absorbed from food, liquids, and administered drugs passes through the walls of the Haversian canals and diffuses into the extracellular fluid. After that, strontium ions are deposited in the bone tissue, where they replace calcium ions by incorporating them into hydroxyapatite crystals [28]. Thus, it provides bone tissue with additional strength and, according to some authors, increases calcium uptake by bone [29]. The tissue content of strontium increases with the administered dose, and its concentration in trabecular tissue is higher than in cortical bone, probably due to the higher metabolism of the latter. After discontinuation, the strontium content of bone tissue decreases rapidly [30].

The effect of strontium on bone depends largely on the dosage, the animal model used, and the kidney function selected for the animal experiment. It has long been known that strontium administered to animals at high doses (8.75 mmol/kg/day) causes significant skeletal abnormalities, mainly when dietary calcium is deficient [31]. In such cases, it contributes to osteomalacia or rickets [32]. Experimental studies on rats with renal insufficiency showed that the administration of strontium at a dose of 150 mg/kg/day causes mineralization disorders resembling osteomalacia [33]. This effect has been associated with inhibiting the activity of 25-hydroxyvitamin D3 alpha-hydroxylase with high doses of strontium [34]. Based on the available literature, low (non-toxic) doses can be considered those that do not exceed 4 mmol Sr/kg/day. Although Grynpas and Marie (1990) indicated that a 0.19% addition of strontium in the diet might cause hypomineralization of bone tissue and reduce the size of hydroxyapatite in young rats, other studies do not confirm this [35]. Biomechanical and histomorphological studies have shown that strontium chloride administered even over a long period has no toxic effect on bone tissue, both in terms of its impact on cells and bone minerals, provided that dietary doses do not exceed one percent [36]. Considering all the information mentioned above, in this experiment, the selection of amounts ensuring both safety for the tested animals and effective incorporation of strontium into the bones was based by the author on the available literature and previous research [18]. The doses of both test substances were adjusted so that both groups of test animals received almost equivalent molar doses of strontium. The animals were fed a standard diet providing adequate amounts of calcium.

In the current study, we decided to test the level of two markers considered benchmarks related to bone tissue metabolism, PINP and CTX I, and additionally a marker of cartilage metabolism, CTX II) [37,38].

PINP is considered one of the most valuable markers for assessing the effects of anabolic and antiresorptive drugs. Its usefulness has been proven in both animal models and cell cultures [39,40]. CTX I is released during the degradation processes of type I collagen (constituting over 90% of the organic bone matrix) during bone tissue resorption. Hence, they are an excellent marker of resorption processes [41,42]. In the literature, the use of strontium ranelate, especially in postmenopausal women, was usually associated with statistically significant changes in bone formation markers and a decrease in bone resorption markers [43,44]. In a study of women with osteoporosis, CTX I levels decreased significantly after one year of treatment with strontium ranelate [45]. However, there have also been studies in which these changes occurred after a long time, and also those in which, despite the observation of the beneficial effect of strontium on bone tissue, significant changes in the level of markers did not occur at all [38,39,46,47].

In the Melatonin–Micronutrients Osteopenia Treatment Study, the authors found that strontium (citrate), combined with melatonin, vitamin D3, and vitamin K2, reduces the level of CTX I and increases the PINP in women in perimenopause after one year of therapy [48]. A study comparing strontium ranelate with teriparatide showed a small but statistically significant decrease in PINP at three months, deepening at six months, and a decrease in CTX I at one month and after two months of medication [49]. In a study combining ranelate with vitamin D in women with osteoporosis, statistically significant differences in PINP levels were obtained after six months of therapy [50]. However, in Rizzoli’s study comparing strontium ranelate with alendronate, no changes in bone markers were noted after 12 months of therapy despite an increase in microarchitectural parameters in women with osteoporosis [46].

In animal studies, changes in marker levels were obtained after long-term therapy. In a study evaluating the long-term effect of strontium ranelate versus zoledronic acid on bone remodeling and bone mineral density (BMD) in ovariectomized rats, PINP levels increased, and CTX I decreased after eight months of therapy [51]. A significant increase in PINP level was also obtained after 12 weeks in a rat osteoporosis model [52].

As an additional marker, we marked degradation products of CTX-II in serum. CTX-II is mainly used in studies assessing the advancement of degenerative changes and rheumatoid arthritis [53,54]. However, its elevated levels are also observed during periods of increased growth, which is associated with increased activity of the growth cartilage. It was also noted that after a period of increased growth, their levels in the serum of the tested animals decreased [55,56].

On the other hand, in our study, there were no statistically significant differences in levels between the control group and the study groups in any of the assessed markers. The lack of observed changes may be due to the short administration period of the tested strontium salts. Also, in the current literature, no studies show how strontium ranelate and chloride administration affect a healthy organism during growth. Our study is the first one using an experimental model, which shows how the level of selected markers of bone turnover changes due to the administration of the tested strontium salts.

Microarchitecture’s influence on bone tissue strength has been repeatedly proven. Although the best way to check the strength of bone tissue is by direct testing, numerous authors confirm that changes in the microarchitecture of bone tissue can be considered its rational predictor [57,58]. In this study, the most favorable area for evaluating the bone tissue microstructure of the examined animals was the distal epiphysis of the femur, rich in trabeculae and free of calcified chondrons. According to numerous authors, this zone has been a recognized location for morphometric studies for many years due to the high content of trabecular and cortical bone, as well as due to the exceptionally fast metabolism of bone tissue in this area, which allows for obtaining the results of research in a relatively short time [59]. Due to the high effectiveness of the microtomographic examination as a tool for assessing the newly formed bone, it was considered the optimal method for assessing the microarchitecture parameters in the applied examination model [60].

In the presented study, the statistical analysis showed a significant effect of the administration of strontium chloride and ranelate on the microarchitectural parameters of the bone trabeculae. An increase in the number of trabeculae, the thickness of the trabeculae, and a decrease in the trabeculae separation were observed, contributing to a significant increase in the percentage of bone tissue. In the known literature, we find evidence indicating the beneficial effect of strontium ranelate on the microstructure of bone tissue. The positive effect of ranelate on the metaphysis of adult female rats was confirmed in the Amman 2004 study. Strontium ranelate administered for 104 weeks increased trabecular thickness and number and decreased trabecular separation in the animal’s proximal tibial metaphysis [61]. Similar results were obtained for subsequent papers by the same author [22] and in the results of Peng and Liu [62]. In adult mice, administration of strontium ranelate increased vertebral body mass [21]. Similar results were obtained in ovariectomized animals. Administration of strontium ranelate to rats for two months immediately after the procedure prevented trabecular bone loss measured via histomorphometry [63].

Histomorphological studies performed on human bone samples also confirmed the beneficial effect of strontium ranelate on trabecular bone microarchitecture. The results of histomorphometric studies of patients participating in studies on the effectiveness of strontium ranelate in women with postmenopausal osteoporosis showed a statistically significant improvement in the parameters studied 1–5 years after the start of treatment [64,65].

In our work, the parameter concerning the cortical bone, informing us about its surface, was also assessed. When our team administered selected strontium salts to animals during growth and development, a significant increase in this parameter was observed. Swiss researchers confirmed that strontium ranelate accelerates the filling of bone defects by quantitatively and qualitatively improving cortical and trabecular bone microarchitecture [66]. That fact may be related to the anabolic effect of strontium on bone tissue remodeling.

In the future, improving the examined parameter may translate into reducing the risk of low-energy fractures associated with the disturbance of the bone tissue microstructure [67]. The changes demonstrated by our team show that regardless of the strontium carrier used, it affects the processes in the bone tissue, which may make it possible to use other strontium salts to promote the formation of a higher peak bone mass.

## 5. Study Limitations

This study has some limitations. The nature of the study is based on limiting animal stress and thus reducing the impact of internal cortisol on bone metabolism. It was achieved, inter alia, by dispensing with the administration of the tested substances using oral gavage in favor of a solution in drinking water. However, the downside of this solution is the possibility of uneven intake of the substance by the tested animals living in one cage. Another limitation results from the single assessment of bone turnover markers at the end of the study without a prior definition of its baseline. The lack of the baseline measurement could have disturbed the correct interpretation of their results.

The standard test for assessing BMD is DEXA (dual-energy X-ray absorptiometry). However, studies have shown that strontium ions are incorporated into hydroxyapatite crystals faster than calcium. Incorporating 1% of strontium into bone has been observed to increase BMD by 10% [23]. As a result, it was confirmed that the administration of strontium overestimates the BMD results obtained in densitometry [68]. Therefore, a conversion factor was established to obtain more reliable BMD results in Sr studies. It should also be noted that there is no analogous conversion factor for BMD tests using micro-CT in the available literature. Consequently, bone mineral density testing using micro-CT cannot be considered fully reliable, contributing to our team not examining BMD. Another limitation of our study is the lack of cell testing. Our goal was to check whether SrCl affects the microarchitecture of constitutive tissue in the same way as SrR since no studies in the existing literature would assess the effects of SrCl on the parameters of bone trabeculae. We should also note that the studied population involved only male mice to avoid gender influence on the results obtained, and no toxicological examination was performed on the animals.

## 6. Conclusions

No studies in the available literature compare strontium chloride to strontium ranelate in a growing mouse model. It should be emphasized that strontium chloride is currently considered a supplement. Our team found only two articles that assessed the effects of this strontium salt on bone tissue. The article by Westberg et al. 2016, a patient case report, showed a positive increase in BMD after two years of strontium chloride. Another study conducted on rats with induced OVX, which compared how strontium from SrCl and SrR salts is incorporated into the bone, showed that Sr in the form of SrR with normal levels of calcium in the diet results in more efficient absorption of strontium into bone tissue compared with SrCl [14,69].

In contrast, strontium ranelate in most countries (except the UK, where it is used as a supplement) is recognized as a drug withdrawn from the market due to side effects. Many studies confirm the SrR-positive effect on bone [17,20,21,22,23,24,25,26,70]. In our work, we have shown that the administration of strontium chloride has a beneficial effect on the microarchitecture of the femur bones of the examined animals, analogously to strontium ranelate. However, we did not observe differences in bone turnover markers, which may be related to the fact that they were only measured at the end of the study. Our results can be considered as preliminary findings for further analyses. Strontium chloride is a promising factor for further studies to determine whether it can be used as an agent to stimulate and maintain peak bone mass.

## Figures and Tables

**Figure 1 nutrients-16-00091-f001:**
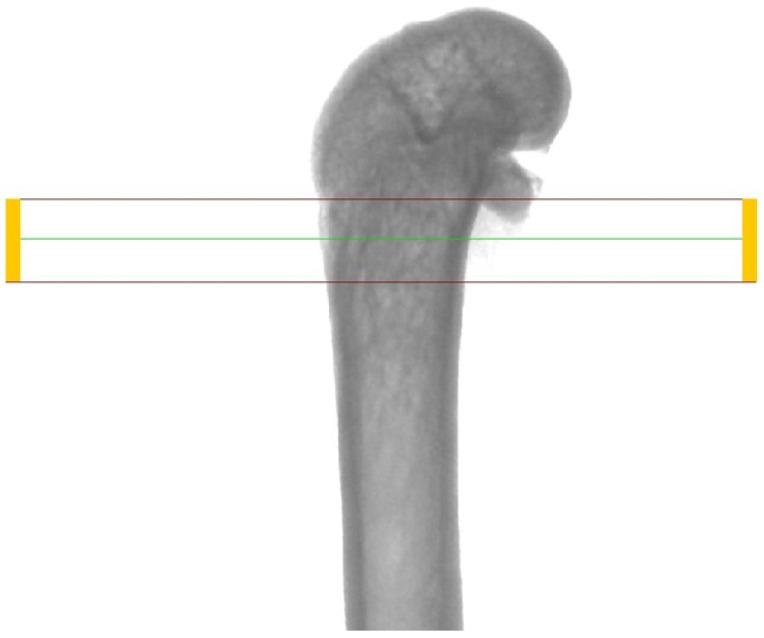
Region of interest (ROI) chosen for investigation (framed)—longitudinal view.

**Figure 2 nutrients-16-00091-f002:**
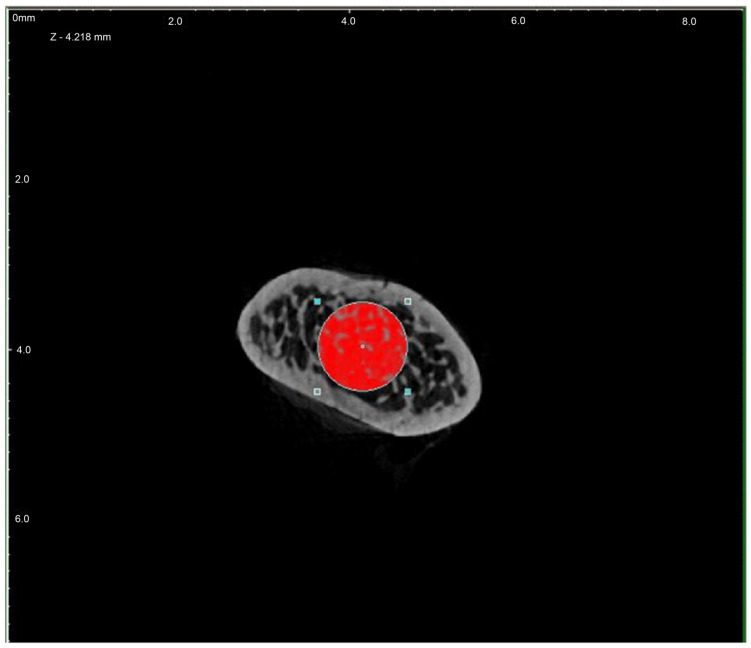
Region of interest (ROI) chosen for investigation—cross-section (red).

**Figure 3 nutrients-16-00091-f003:**
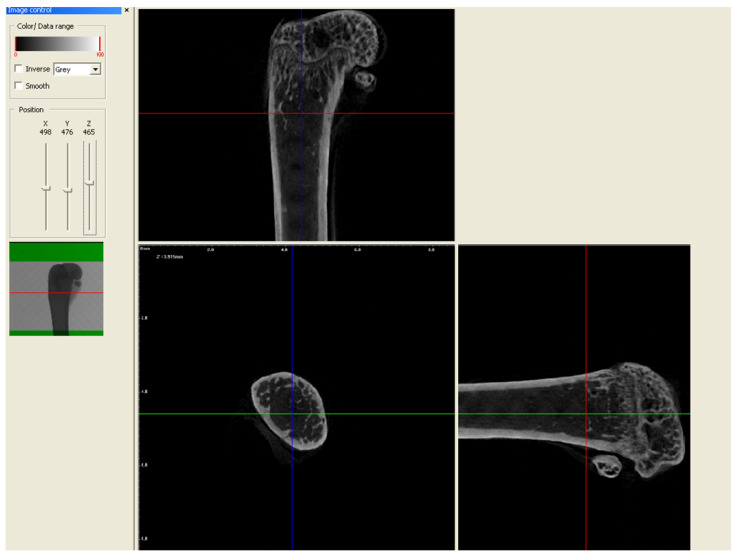
Representative overview photos for group control. The picture was taken during the micro-CT analysis (DataViewer SKYSCAN).

**Figure 4 nutrients-16-00091-f004:**
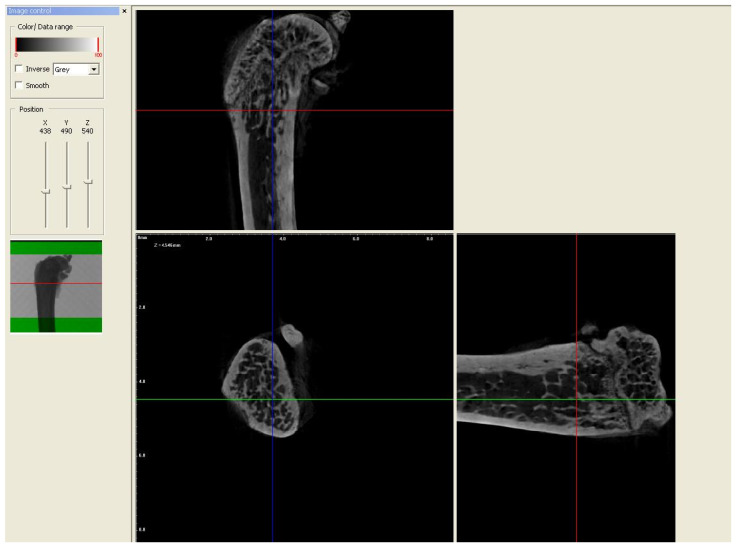
Representative overview photos for group strontium chloride. The picture was taken during the micro-CT analysis (DataViewer SKYSCAN).

**Figure 5 nutrients-16-00091-f005:**
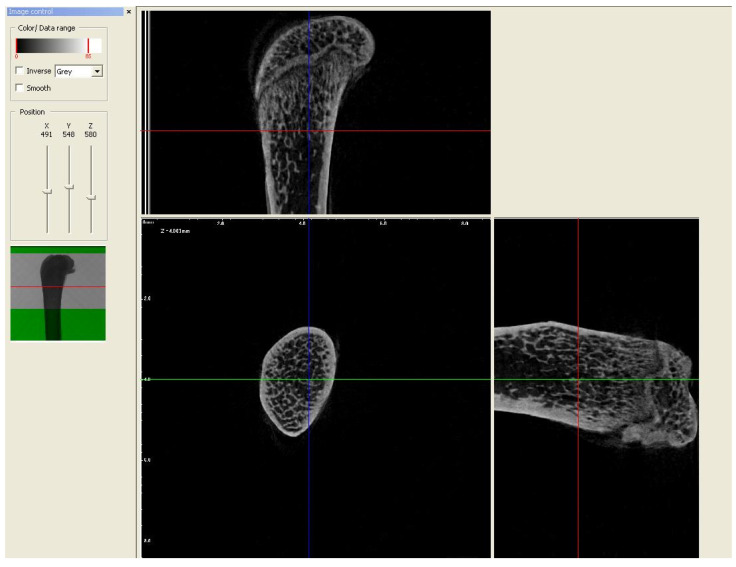
Representative overview photos for group strontium ranelate. The picture was taken during the micro-CT analysis (DataViewer SKYSCAN).

**Table 1 nutrients-16-00091-t001:** Weight gain between the first and last day of the experiment.

Group	Weight Gain (g)
Control	11.993 ± 1.028
Strontium chloride	11.673 ± 1.653
Strontium ranelate	11.674 ± 1.268

Values represent mean ± SEM. The sample size for each group is *n* = 14. No significance was found between any group.

**Table 2 nutrients-16-00091-t002:** Femur micro-CT-analyzed trabecular thickness (Tb.Th) (µm) outcomes of growing SWISS mice administrated with strontium chloride and strontium ranelate.

Group	N	Tb.Th (µm) Mean	Tb.Th (µm) Std.Dev.	Tb. Th (µm) Std.Err
Control	14	6.289619	1.605175	0.429001
Strontium chloride	14	8.033544 *	1.837590	0.491117
Strontium ranelate	14	8.665711 **	1.869460	0.499634

The sample size for each group is *n* = 14. Where significance in one-way ANOVA was found, post hoc analysis was performed. Group comparisons are indicated by superscript. The significant differences per the Tukey–Kramer test were as follows: between the control and strontium chloride groups, * (*p* < 0.05); between the control and strontium ranelate groups, ** (*p* < 0.02). Std.Dev—standard deviation; Std.Err—standard error.

**Table 3 nutrients-16-00091-t003:** Post hoc Tukey–Kramer test for trabecular thickness (Tb.Th) (µm).

	Tukey HSD Test; Variable Tb.Th (µm) Approximate Probabilities for Post Hoc Tests Error: Between MS = 3.1494, df = 39.000			
	Group	(1) 6.2896	(2) 8.0335	(3) 8.6657
1	Control		0.034361 *	0.003032 **
2	Strontium chloride	0.034361 *		0.617156
3	Strontium ranelate	0.003032 **	0.617156	

Significant differences: * (*p* < 0.05); ** (*p* < 0.02).

**Table 4 nutrients-16-00091-t004:** Femur micro-CT-analyzed trabecular number (Tb.N) (1/mm) outcomes of growing SWISS mice administrated with strontium chloride and strontium ranelate.

Group	N	Tb.N (1/mm) Mean	Tb.N (1/mm) Std.Dev.	Tb.N (1/mm) Std.Err
Control	14	0.008873	0.008141	0.002176
Strontium chloride	14	0.022479 **	0.009416	0.002517
Strontium ranelate	14	0.027754 ***	0.012664	0.003385

The sample size for each group is *n* = 14. Where significance in one-way ANOVA was found, post hoc analysis was performed. Group comparisons are indicated by superscript. The significant differences per the Tukey–Kramer test were as follows: between the control and strontium chloride groups, ** (*p* < 0.02); between the control and strontium ranelate groups, *** (*p* < 0.001). Std.Dev—standard deviation; Std.Err—standard error.

**Table 5 nutrients-16-00091-t005:** Post hoc Tukey–Kramer test for trabecular number (Tb. N) (1/mm).

	Tukey HSD Test; Variable Tb.N (1/mm) Approximate Probabilities for Post Hoc Tests Error: Between MS = 0.00011, df = 39.000			
	Group	(1) 0.00887	(2) 0.02248	(3) 0.02775
1	Control		0.003299 **	0.000170 ***
2	Strontium chloride	0.003299 **		0.371000
3	Strontium ranelate	0.000170 ***	0.371000	

Significant differences: ** (*p* < 0.02); *** (*p* < 0.001).

**Table 6 nutrients-16-00091-t006:** Femur micro-CT-analyzed percent of bone volume (BV/TV) (%) outcomes of growing SWISS mice administrated with strontium chloride and strontium ranelate.

Group	N	BV/TV (%) Mean	BV/TV (%) Std.Dev.	BV/TV (%) Std.Err
Control	14	6.16186	5.74342	1.534992
Strontium chloride	14	18.54775 **	10.27522	2.746167
Strontium ranelate	14	23.70778 ***	11.46758	3.064840

The sample size for each group is *n* = 14. Where significance in one-way ANOVA was found, post hoc analysis was performed. Group comparisons are indicated by superscript. The significant differences per the Tukey–Kramer test were as follows: between the control and strontium chloride groups, ** (*p* < 0.02); between the control and strontium ranelate groups, *** (*p* < 0.001). Std.Dev—standard deviation; Std.Err—standard error.

**Table 7 nutrients-16-00091-t007:** Post hoc Tukey–Kramer test for percent of bone volume (BV/TV) (%).

	Tukey HSD Test; Variable BV/TV (%) Approximate Probabilities for Post Hoc Tests Error: Between MS = 90.024, df = 39.000			
	Group	(1) 6.1619	(2) 18.548	(3) 23.708
1	Control		0.003861 **	0.000167 ***
2	Strontium chloride	0.003861 **		0.331406
3	Strontium ranelate	0.000167 ***	0.331406	

Significant differences: ** (*p* < 0.02); *** (*p* < 0.001).

**Table 8 nutrients-16-00091-t008:** Femur micro-CT-analyzed trabecular separation (Tb. Sp) (µm) outcomes of growing SWISS mice administrated with strontium chloride and strontium ranelate.

Group	N	Tb.Sp (µm) Mean	Tb.Sp (µm) Std.Dev.	Tb.Sp (µm) Std.Err
Control	14	37.971	13.366	0.068
Strontium chloride	14	25.119 *	9.075	0.046
Strontium ranelate	14	21.255 **	7.36	0.037

The sample size for each group is *n* = 14. The significant differences are as follows: between the control and strontium chloride groups, * (*p* < 0.05); between the control and strontium ranelate groups, ** (*p* < 0.02); and no significant difference was found between the strontium chloride and strontium ranelate groups. Std.Dev.—standard deviation; Std.Err—standard error.

**Table 9 nutrients-16-00091-t009:** Kruskal–Wallis test for trabecular separation (Tb. Sp) (µm).

	Multiple Comparisons *p* Values (2-Tailed); Tb.Sp. (µm) Independent (Grouping) Variable: Group Kruskal–Wallis test: H (2, N = 42) = 11.44850 *p* = 0.0033		
	Control R: 30.357	Strontium Chloride R: 18.714	Strontium Ranelate R: 15.429
Control		0.036121 *	0.003851 **
Strontium chloride	0.036121 *		1.000000
Strontium ranelate	0.003851 **	1.000000	

Significant differences: * (*p* < 0.05); ** (*p* < 0.02).

**Table 10 nutrients-16-00091-t010:** Femur micro-CT-analyzed cortical area (µm^2^) outcomes of growing SWISS mice administrated with strontium chloride and strontium ranelate.

Group	N	Cortical Area (µm^2^) Mean	Cortical Area (µm^2^) Std.Dev.	Cortical Area (µm^2^) Std.Err
Control	14	14,802.05	2139.508	571.807
Strontium chloride	14	19,856.42 ***	4972.380	1328.924
Strontium ranelate	14	19,246.10 **	2496.255	667.152

The sample size for each group is *n* = 14. Where significance in one-way ANOVA was found, post hoc analysis was performed. Group comparisons are indicated by superscript. The significant differences per the Tukey–Kramer test were as follows: between the control and strontium chloride groups, *** (*p* < 0.001); between the control and strontium ranelate groups, ** (*p* < 0.02). Std.Dev.—standard deviation; Std.Err—standard error.

**Table 11 nutrients-16-00091-t011:** Post hoc Tukey–Kramer test for the cortical area (µm^2^).

	Tukey HSD Test; Variable Cortical Area (µm^2^) Approximate Probabilities for Post Hoc Tests Error: Between MS = 1184 × 10^4^ df = 39.000			
	Group	(1) 14,802.	(2) 19,856.	(3) 19,246.
1	Control		0.001199 ***	0.004270 **
2	Strontium chloride	0.001199 ***		0.886159
3	Strontium ranelate	0.004270 **	0.886159	

Significant differences: ** (*p* < 0.02); *** (*p* < 0.001).

**Table 12 nutrients-16-00091-t012:** Serum procollagen type I N-terminal propeptide (PINP) (µg/L) outcomes of growing SWISS mice administrated with strontium chloride and strontium ranelate.

Group	N	PINP (µg/L) Mean	PINP (µg/L) Std.Dev.	PINP (µg/L) Std.Err
Control	9	8.679778	1.167847	0.389282
Strontium chloride	9	9.643667	2.670709	0.890236
Strontium ranelate	9	7.850000	1.582566	0.527522

The sample size for each group is *n* = 9. No statistical difference was found. Std.Dev.—standard deviation; Std.Err—standard error.

**Table 13 nutrients-16-00091-t013:** Serum carboxy-terminal collagen crosslinks I (CTXI) (µg/L) outcomes of growing SWISS mice administrated with strontium chloride and strontium ranelate.

Group	N	CTXI (µg/L) Mean	CTXI (µg/L) Std.Dev.	CTXI (µg/L) Std.Err
Control	9	21.95556	4.496855	1.498952
Strontium chloride	9	19.32400	5.389408	1.796469
Strontium ranelate	9	20.06567	8.385476	2.795159

The sample size for each group is *n* = 9. No statistical difference was found. Std.Dev.—standard deviation; Std.Err—standard error.

**Table 14 nutrients-16-00091-t014:** Serum carboxy-terminal collagen crosslinks II (CTXII) (µg/L) outcomes of growing SWISS mice administrated with strontium chloride and strontium ranelate.

Group	N	CTXII (µg/L) Mean	CTXII (µg/L) Std.Dev.	CTXII (µg/L) Std.Err
Control	9	153.8659	76.30417	25.43472
Strontium chloride	9	145.1188	72.77673	24.25891
Strontium ranelate	9	126.4287	58.02944	19.34315

The sample size for each group is *n* = 9. No statistical difference was found. Std.Dev.—standard deviation; Std.Err—standard error.

## Data Availability

Data are contained within the article.

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
