# Peer review of "Strontium Ranelate and Strontium Chloride Supplementation Influence on Bone Microarchitecture and Bone Turnover Markers—A Preliminary Study"

_nutrients, 2023, doi:10.3390/nu16010091_

Round 1
Reviewer 1 Report (Previous Reviewer 2)
Comments and Suggestions for Authors
In entire manuscript, there are not sufficient data or evidences to support the conclusion.
Comments on the Quality of English LanguageEnglish language editing required
Author Response
In entire manuscript, there are not sufficient data or evidences to support the conclusion.
Response: Thank you very much for your review; the conclusions have been re-edited.
Reviewer 2 Report (New Reviewer)
Comments and Suggestions for Authors
The authors of the manuscript “Strontium ranelate and strontium chloride supplementation influence on the bone microarchitecture and bone turnover markers” demonstrated in a mouse model that organic and inorganic strontium compounds have similar effect on the bone microarchitecture and strength. Some suggestions to improve the paper:
· Some results are negative, is the sample size adequate to demonstrate that there are no differences between Strontium ranelate and strontium chloride ssupplementation. How was the sample size calculated?
· Among the limitations of the study it should be noted that the population studied involved only male mice
· Please add to the methods section the intra and inter-assay variability of the kits used for P1NP and CTx measurements
Author Response
Thank you very much for your review. All comments allow us to introduce valuable changes to our manuscript.
All responses to individual comments are presented below.
- Some results are negative, is the sample size adequate to demonstrate that there are no differences between Strontium ranelate and strontium chloride supplementation. How was the sample size calculated?
Response: Following the reduction principle, the number of animals used in the experiment was kept to a minimum. For this purpose, a test power analysis, which allows for the assessment of the minimum number of animals used in one research group, was conducted using the Statistica statistical program. The analysis showed that groups should consist of no less than 14 individuals to enable a correct, reliable, and trustworthy statistical analysis of the results obtained during the experiment (lines: 92-94).
In the case of ELISA kits, the determination of selected bone turnover markers was performed on a smaller number of individuals because some of the samples undergo hemolysis during centrifugation so that the statistical analysis was reliable, the Tukey test could be performed, and the tested groups were homogeneous; the number of samples was reduced to the same in all groups (lines: 154-156).
- Among the limitations of the study, it should be noted that the population studied involved only male mice.
Response: Added – lines: 458-459.
- Please add to the methods section the intra and inter-assay variability of the kits used for P1NP and CTx measurements.
Response: Added – lines: 156-159.
Reviewer 3 Report (New Reviewer)
Comments and Suggestions for Authors
The manuscript of Turżańska et al. addresses an interesting study about the potential influence of supplementation with strontium chlorate and strontium ranelate on the mice bone microarchitecture.
In general, the manuscript fulfills the requirements of Nutrients Journal. The title is specific, and the abstract summarizes the most important findings in 163 words (the maximum is established in 200 words) in a single paragraph. Introduction section includes the objective of the study. The manuscript provides more than 3 but less than 10 keywords (concretely 5). Methodologies section is well-explained. The present study was conducted in accordance with the Declaration of Helsinki, and it was approved by the Ethics Committee of Medical University of Lublin (KE 35 425/2015). Regarding results, this work provides interesting findings for future studies. References are correctly identified in the text and properly present in the references section.
However, important modifications must be done to consider the publication of this study. The specific comments and suggestions for improving the quality of the manuscript are the following:
(1) A concise cover letter should have been submitted with the manuscript to explain originality of the findings of the study and the reasons for why the work fits the scope of Nutrients Journal and why should it be published.
(2) The title of the manuscript must be adequately adjusted. It should be indicated that this work is a preliminary study performed in mice to make the difference with clinical evidence in humans and to avoid any potential misunderstanding of readers.
(3) A more complete and specific background should be included in the Introduction section to better address the approach of this study. Please consider including the most recent references.
(4) Line 125: General observations. In this section, authors reported that no deaths were recorded, none of the animals were visibly ill during the experiment, and no anomalies were noted during the macroscopic autopsy of the animals. It seems that no toxicological exams were done in mice. If not, it would be an important critical point to include in section 5 “Study Limitations” (line 385).
(5) Conclusions section must be rewritten. Authors must eliminate any categorical statement such as “Strontium ranelate is a substance with a proven effect on microarchitectural parameters and bone tissue strength. Therefore, we can consider it as a reference substance” (lines 405-406) and “In light of the presented results, it seems that strontium chloride, as an easily available, simple chemical compound, may be a promising component of supplements supporting the achievement and maintenance of optimal bone mass” (lines 414-416). Please notice that the present work is a preliminary study in mice, so that evidence found is insufficient to establish cause-effect relationships and to make these assumptions.
(6) References section must be checked and updated. Please notice that several references are not recent: reference 5 (1981), reference 11 (1996), reference 20 (1962), reference 23 (1995), reference 25 (1994) ...
Minor editing of English language required.
Author Response
Thank you for your review. All comments have been incorporated into the revised manuscript. Our work has been supplemented with all comments:
All responses to individual comments are presented below.
- A concise cover letter should have been submitted with the manuscript to explain originality of the findings of the study and the reasons for why the work fits the scope of Nutrients Journal and why should it be published.
Response: The cover letter has been changed and added as an attachment.
- The title of the manuscript must be adequately adjusted. It should be indicated that this work is a preliminary study performed in mice to make the difference with clinical evidence in humans and to avoid any potential misunderstanding of readers.
Response: The title has been changed.
- A more complete and specific background should be included in the Introduction section to better address the approach of this study. Please consider including the most recent references.
Response: The introduction has been expanded, and more recent references have been added.
- Line 125: General observations. In this section, authors reported that no deaths were recorded, none of the animals were visibly ill during the experiment, and no anomalies were noted during the macroscopic autopsy of the animals. It seems that no toxicological exams were done in mice. If not, it would be an important critical point to include in section 5 “Study Limitations” (line 385).
Response: Added – lines: 457-458.
- Conclusions section must be rewritten. Authors must eliminate any categorical statement such as “Strontium ranelate is a substance with a proven effect on microarchitectural parameters and bone tissue strength. Therefore, we can consider it as a reference substance”(lines 405-406) and “In light of the presented results, it seems that strontium chloride, as an easily available, simple chemical compound, may be a promising component of supplements supporting the achievement and maintenance of optimal bone mass” (lines 414-416). Please notice that the present work is a preliminary study in mice, so that evidence found is insufficient to establish cause-effect relationships and to make these assumptions.
Response: The conclusions section has been rewritten (lines:474-484).
- References section must be checked and updated. Please notice that several references are not recent: reference 5 (1981), reference 11 (1996), reference 20 (1962), reference 23 (1995), reference 25 (1994).
Response: References have been checked and updated.

Round 2
Reviewer 1 Report (Previous Reviewer 2)
Comments and Suggestions for Authors
Comments (v2) for nutrients-2687186
1. The English of main text must be edited further. For example, there are the first using the “According to the…” in 40 lines, and second using the “According to…” in the same paragraph, etc.
2. The table must use a three-line grid.
3. The font size in the picture has slightly increased to make it clear for readers to read.
4. As previous mentioned, there are not sufficient data or evidences to support the conclusion in entire manuscript. For example, some necessary parameters between bone transitions, such as changes in osteoblasts, osteoclasts, osteoblasts, chondrocytes, etc.
5. Therefore, I still recommend rejecting the manuscript.
Comments on the Quality of English LanguageIt is recommended that scientific papers must use some scientific language and logic.
Author Response
Dear Reviewer,
Thank you for your kind comments and reviews, which helped us to improve our manuscript. All comments have been incorporated in the revised manuscript.
Detailed comments:
- The English of main text must be edited further. For example, there are the first using the “According to the…” in 40 lines, and second using the “According to…” in the same paragraph, etc.
Response:
Thank you for your comments. The language and style have been improved.
- The table must use a three-line grid.
Response:
Thank you for the remark. Table 1 has been corrected.
- The font size in the picture has slightly increased to make it clear for readers to read
Response:
The photos used are screenshots from the actual micro-CT examinations. We aimed to present how the examined animals' femurs were analyzed graphically. We have enlarged them to make them more readable. Unfortunately, we cannot interfere with the font format of the study screenshot without disturbing its authenticity.
- As previous mentioned, there are not sufficient data or evidences to support the conclusion in entire manuscript. For example, some necessary parameters between bone transitions, such as changes in osteoblasts, osteoclasts, osteoblasts, chondrocytes, etc.
Response:
One of the limitations of our study is the lack of cell testing. Our goal was to check whether SrCl affects the microarchitecture of constitutive tissue in the same way as SrR. No studies in the existing literature would assess the effects of SrCl on the parameters of bone trabeculae. Strontium chloride is a dietary supplement that can be purchased without a prescription. Our team found only two articles that assessed the effects of this strontium salt on bone tissue. The article by Westberg et al. 2016, a patient case report, showed a positive increase in BMD after two years of using this supplement. Another study conducted on rats with induced OVX, which compared how strontium from SrCl and SrR salts is incorporated into the bone, showed that Sr in the form of SrR with normal levels of calcium in the diet results in more efficient absorption of strontium into bone tissue compared to SrCl.
The results obtained by our team are new because they concern the impact of SrCl administration on the microarchitecture of the trabecular bone structure and the level of selected bone turnover markers (CTX-I, CTX-II, PINP). These results complement the available knowledge.
The results of our work confirm that SCl has the same effect as SrR on the microarchitecture of bone tissue and allow for further, more detailed research. Future work will examine cellular activity in bone and cartilage tissues, including osteoclasts, osteocytes, and chondrocytes.
Literature:
Westberg, S.M.; Awker, A.; Torkelson, C.J. Use of Strontium Chloride for the Treatment of Osteoporosis: A Case Report. Altern Ther Health Med. 2016,, 22(3), 66-70.
Pemmer B, Hofstaetter JG, Meirer F, Smolek S, Wobrauschek P, Simon R, Fuchs RK, Allen MR, Condon KW, Reinwald S, Phipps RJ, Burr DB, Paschalis EP, Klaushofer K, Streli C, Roschger P. Increased strontium uptake in trabecular bone of ovariectomized calcium-deficient rats treated with strontium ranelate or strontium chloride. J Synchrotron Radiat. 2011 Nov;18(Pt 6):835-41. doi: 10.1107/S090904951103038X. Epub 2011 Sep 15. PMID: 21997907.
- Therefore, I still recommend rejecting the manuscript.
Response:
Thank you for your comments. We improved the manuscript due to your kind remarks. We rewrote the highlighted parts as you suggested to clarify our idea of the manuscript content. We hope that, thanks to your kind remarks, this form of the manuscript will be interesting for readers.
Reviewer 3 Report (New Reviewer)
Comments and Suggestions for Authors
Dear Authors,
After carefully reviewing the last version of the manuscript, I consider it is worthy of publication. The suggestions I proposed have been addressed and the quality of the manuscript has increased.
Sincerely
Author Response
Dear Reviewer,
Thank you for all your help and support.
Sincerely,
Authors
This manuscript is a resubmission of an earlier submission. The following is a list of the peer review reports and author responses from that submission.
Round 1
Reviewer 1 Report
Comments and Suggestions for Authors
The authors of this manuscript compared the effects of strontium chlorate and strontium ranelate on bone microarchitecture in mice. The study is very interesting, and the manuscript is well-written. However, the work has some shortcomings that should be filled and some errors to be corrected. I think this article may be eligible for publication after the following major revisions have been made.
-The cortical area was examined as a whole, but why for the trabecular bone only the area corresponding to the red circle was assessed? Based on which criterion was only this area of interest selected?
-The study assessed sufficient bone parameters considering that it is a predominantly morphological study, however, because the software will surely have automatically generated also Bone Mineral Density (BMD) data, why wasn't data on this parameter included? Adding a graph on BMD could certainly add value to the manuscript considering that it is a parameter that reflects the state of health of the bone and therefore could be a confirmatory element to support all the other data.
-The bones in Figure 2 on page 8 should have had the same orientation in all images.
-In the materials and methods section, enter information about the number of slides considered to create Figure 1a.
-In Graph 5, all characters (letters and numbers) are larger than in the other graphs. Reformat all graphs so that they look the same.
-All graphs should show the units of the Cartesian axes.
-It would be appropriate to remove the comma that appears immediately after the numbers in some tables.
-All graphs have an inappropriate title. For example “Graph. 1 One-way ANOVA for body mass gain."Correct this by first writing a title that is explanatory about the content of the graph and then enter the information about the statistical analysis.
-CTX and PINP markers on what type of sample were they assessed? On blood? On serum? Please specify in the text.
-Always indicate towards which group significance is intended.
-Write the discussion as a single paragraph, see template. The materials and methods section, on the other hand, should be divided into paragraphs.
-In table 8, the symbol # is not shown.
-Indicate significance in graphs.
-Indicate the SEM in all graphs and tables.
-Authors wrote 'Random overview' images, are they representative images or randomized images? How was the random choice made?
-The discussion on bone turnover markers is too elementary. Although the figure is not significant, it would still be appropriate to better explain the trend of these markers by correlating it with the other results obtained.
Comments on the Quality of English LanguageMinor editing of English language required
Author Response
Thank you very much for your valuable comments and pointing out errors. We hope that all noticed errors have been corrected.
- The cortical area was examined as a whole, but why for the trabecular bone only the area corresponding to the red circle was assessed? Based on which criterion was only this area of interest selected?
Response: The study focused mainly on trabecular bone as the most susceptible to changes and more metabolically active. The ROI was a cylinder starting just below the visually identified growth plate and covering 100 slices. The ROI diameter was matched to the internal dimension of the smallest of the examined bones so that it did not cover the cortical bone and was identical in all tested samples. The distal epiphysis of the femur was chosen for the examination as the most favorable area for assessing the bone tissue microstructure. The chosen area is rich in trabeculae, free from calcified chondrons. According to many authors, this zone has been a recognized location for morphometric studies for many years due to the high content of trabecular and cortical bone, as well as due to the exceptionally fast metabolism of bone tissue in this area, which allows for obtaining the results of research in a relatively short time.
- The study assessed sufficient bone parameters considering that it is a predominantly morphological study; however, because the software will surely have automatically generated also Bone Mineral Density (BMD) data, why wasn't data on this parameter included? Adding a graph on BMD could certainly add value to the manuscript considering that it is a parameter that reflects the state of health of the bone and therefore could be a confirmatory element to support all the other data.
Response: Thank you for this comment; it is very valid and helpful. Unfortunately, our version of the software does not have this feature. We will fix this in further research.
- The bones in Figure 2 on page 8 should have had the same orientation in all images.
Response: Thank you for the observation. It has been corrected.
- In the materials and methods section, enter information about the number of slides considered to create Figure 1a.
Response: It was 100 slides. The information has been added to the text. Line: 98.
- In Graph 5, all characters (letters and numbers) are larger than in the other graphs. Reformat all graphs so that they look the same.
Response: All of the graphs and figures have been corrected as suggested.
- All graphs should show the units of the Cartesian axes.
Response: All of the graphs and figures have been corrected as suggested.
- It would be appropriate to remove the comma that appears immediately after the numbers in some tables.
Response: All commas have been removed and replaced with dots.
- All graphs have an inappropriate title. For example “Graph. 1 One-way ANOVA for body mass gain."Correct this by first writing a title that is explanatory about the content of the graph and then enter the information about the statistical analysis.
Response: All graph descriptions have been changed.
- CTX and PINP markers on what type of sample were they assessed? On blood? On serum? Please specify in the text.
Response: The level of bone turnover markers has been assessed in the blood serum collected from the examined animals. The note has been included in the materials and methods section. Lines: 138-140.
- Always indicate towards which group significance is intended.
Response: All of the tables have been corrected.
- Write the discussion as a single paragraph; see template. The materials and methods section, on the other hand, should be divided into paragraphs.
Response: The discussion has been rewritten in the form of one paragraph. The materials and methods section has been divided into paragraphs.
- In Table 8, the symbol # is not shown.
Response: The error has been corrected in the table description. There was no significant difference between strontium chloride and strontium ranelate groups, so the symbol # does not appear in the table.
- Indicate significance in graphs.
Response: Statistical significance symbols have been placed on all graphs.
- Indicate the SEM in all graphs and tables.
Response: The SEM has been included in all tables and graphs.
- Authors wrote 'Random overview' images; are they representative images or randomized images? How was the random choice made?
Response: This is our mistake; it is a representative image. The description of Figure 2 has been corrected. Lines: 133-134.
-The discussion on bone turnover markers is too elementary. Although the figure is not significant, it would still be appropriate to better explain the trend of these markers by correlating it with the other results obtained.
Response: The bone turnover markers part has been added to the discussion. Lines: 464-511.
Reviewer 2 Report
Comments and Suggestions for Authors
Comments to manuscript ID nutrients-2542306
The authors have presented a relatively monotonous research that one single method to measure the bone microstructures of SWISS growing mice. This seems to contradict the basic demands of the journal and the strictness of scientific study. However, some suggestions / corrections must be revised.
1. All English language should be modified. Some spell or format errors should be corrected, e.g. one extra space should be deleted in line 23, “( )” should be changed to “[ ]” in line 66 and 67, etc.
2. What is the BV/TV? Please write its correct name not bone volume in line 25. “TbTh” , “TbN”should be changed to “Tb.Th”, “Tb.N”, respectively. Tb.N is the full name that is Trabecular number. There are other places of the text that need to be modified carefully.
3. What are the breeds of cows?
4. In the sections of introduction and discussion, all references should be managed base on your current manuscript. There is more information from published papers that not fully presented or they may not have grasped the author's true meaning. Thus, the manuscript must be re-edited.
5. In entire manuscript, there are not sufficient data or evidences to support the conclusion.
Comments on the Quality of English Language
All English language should be modified.
Author Response
Thank you very much for your valuable comments and pointing out errors. We hope that all noticed errors have been corrected.
- All English language should be modified. Some spell or format errors should be corrected, e.g. one extra space should be deleted in line 23, “( )” should be changed to “[ ]” in line 66 and 67, etc.
Response: All of the paper has been re-edited. A native speaker has checked English.
- What is the BV/TV? Please write its correct name not bone volume in line 25. “TbTh” , “TbN”should be changed to “Tb.Th”, “Tb.N”, respectively. Tb.N is the full name that is Trabecular number. There are other places of the text that need to be modified carefully.
Response: All mistakes have been corrected.
- What are the breeds of cows?
Response: The experiment was carried out on growing SWISS mice.
- In the sections of introduction and discussion, all references should be managed base on your current manuscript. There is more information from published papers that not fully presented or they may not have grasped the author's true meaning. Thus, the manuscript must be re-edited.
Response: Our team has double-checked all of the references. All of the manuscript has been checked and rewritten.
- In entire manuscript, there are not sufficient data or evidences to support the conclusion.
Response: Results and discussion has been redrafted to confirm our conclusion. Our studies have shown that the administration of the inorganic strontium salt SrCl to growing male mice positively affects the parameters of bone tissue microarchitecture. The results are not significantly different from the clinically proven reference substance - SrR.
Round 2
Reviewer 2 Report
Comments and Suggestions for Authors
1, The format of some articles must be corrected, for example: "a." and "b." are redundant in line 144, please delete it; ",' is redundant in table 1, please delete it; "p" should be changed to the "P" in statistics in manuscript; etc.
2, Several table data was modified in the second version compared to the first version, for example: "Table 3. Post Hoc Tukey-Kramer Test for Trabecular Thickness (Tb.Th) [μm] ", the author added the different significant difference in the second manuscript, there is different diversity indices that is P<0.02, P<0.001. However, the best way is to unify the statistics of the entire text in section of 2.4.
3, The results of the table and the graph are duplicated, and one of them must be left.
4, Shorten the discussion and conclusion to better clarify the central meaning of this article.
Comments on the Quality of English LanguageExtensive editing of English language required and reduces redundant sentences in the article.
Author Response
Thank you for all of the valuable and constructive comments. We hope that all noticed errors have been corrected.
1, The format of some articles must be corrected, for example: "a." and "b." are redundant in line 144, please delete it; ",' is redundant in table 1, please delete it; "p" should be changed to the "P" in statistics in manuscript; etc.
All of the identified mistakes and errors have been corrected.
2, Several table data was modified in the second version compared to the first version, for example: "Table 3. Post Hoc Tukey-Kramer Test for Trabecular Thickness (Tb.Th) [μm] ", the author added the different significant difference in the second manuscript, there is different diversity indices that is P<0.02, P<0.001. However, the best way is to unify the statistics of the entire text in section of 2.4.
In the second corrected version, the levels of statistical difference have been added to clarify the results. According to the current suggestion, we unify the statistics in the 2.4 section – lines 141-142.
3, The results of the table and the graph are duplicated, and one of them must be left.
All of the graphs have been removed.
4, Shorten the discussion and conclusion to better clarify the central meaning of this article.
The discussion and conclusions have been shortened.